# Black Ginseng Ameliorates Cellular Senescence via p53-p21/p16 Pathway in Aged Mice

**DOI:** 10.3390/biology11081108

**Published:** 2022-07-25

**Authors:** Su-Jeong Lee, Da-Yeon Lee, Jennifer F. O’Connell, Josephine M. Egan, Yoo Kim

**Affiliations:** 1Department of Nutritional Sciences, Oklahoma State University, Stillwater, OK 74078, USA; crystal.lee10@okstate.edu (S.-J.L.); dayeon.lee@okstate.edu (D.-Y.L.); 2Laboratory of Clinical Investigation, National Institute on Aging (NIA), Baltimore, MD 21224, USA; jennifer.oconnell@nih.gov (J.F.O.); eganj@grc.nia.nih.gov (J.M.E.)

**Keywords:** black ginseng, cellular senescence, senolytics, complement C1q, β-catenin, p53, p21, p16

## Abstract

**Simple Summary:**

The goal of this study was to examine if BG impacts the aging process, specifically cellular senescence, using in vitro and aged mouse models. Primary mouse embryonic fibroblasts (MEFs) and aged mice (18 months old) showed that BG supplementation retarded cellular senescence. Of note, BG-supplemented aged mice had remarkedly altered hepatic genes involved in the aging process as it caused less activation of the canonical senescence pathway. These observations demonstrated that BG positively impacts the age-related phenotype by controlling the expression of cellular senescence in the liver and other metabolic organs such as skeletal muscle and white adipose tissue.

**Abstract:**

Cellular senescence, one of the hallmarks of aging, refers to permanent cell cycle arrest and is accelerated during the aging process. Black ginseng (BG), prepared by a repeated steaming and drying process nine times from fresh ginseng (*Panax ginseng* C.A. Meyer), is garnering attention for herbal medicine due to its physiological benefits against reactive oxygen species (ROS), inflammation, and oncogenesis, which are common cues to induce aging. However, which key nodules in the cellular senescence process are regulated by BG supplementation has not been elucidated yet. In this study, we investigated the effects of BG on cellular senescence using in vitro and aged mouse models. BG-treated primary mouse embryonic fibroblasts (MEFs) in which senescence was triggered by ionizing radiation (IR) expressed less senescence-associated β-galactosidase (SA-β-gal)-positive stained cells. In our aged mice (18 months old) study, BG supplementation (300 mg/kg) for 4 weeks altered hepatic genes involved in the aging process. Furthermore, we found BG supplementation downregulated age-related inflammatory genes, especially in the complement system. Based on this observation, we demonstrated that BG supplementation led to less activation of the canonical senescence pathway, p53-dependent p21 and p16, in multiple metabolic organs such as liver, skeletal muscle and white adipose tissue. Thus, we suggest that BG is a potential senolytic candidate that retards cellular senescence.

## 1. Introduction

Ginseng, a root of a plant from the genus *Panax*, has been widely used as conventional herbal medicine in East Asia for millennia because of its putative medicinal values [1]. Raw ginseng is processed into white and red ginseng by steaming and drying procedures [2]. Among the processed ginseng products, black ginseng (BG) is prepared from Korean ginseng (*Panax ginseng*) by a repeated steaming and drying process nine times and turns into a black color due to the Maillard reaction [3]. During this process, the biological and pharmacological activities of BG are enhanced, including ginsenoside contents, phenolic compounds, Maillard reaction products, and antioxidant activities, compared to white or red ginseng [4,5,6]. Recent studies have revealed that BG has ameliorative effects on age-related functional decline such as anti-inflammatory and antioxidant effects [5,7,8]. However, a specific molecular mechanism by which BG regulates the aging-associated processes largely remains unclear.

Cellular senescence refers to a stable and mostly permanent cell-cycle arrest characterized by the accumulation of severe cellular damage caused by ultraviolet (UV), ionizing radiation (IR), reactive oxygen species (ROS), and replication errors [9,10]. It has emerged as an important hallmark of aging and age-associated diseases such as obesity, type 2 diabetes, cardiovascular disease, and cancer because senescent cells accumulate exponentially in aged tissues [11,12,13]. These cells secrete proinflammatory cytokines, chemokines, and growth factors that are hallmarks of the senescence-associated secretory phenotypes (SASPs), which affect tissue structure and functions, inhibit the proper functioning of the immune system, and can cause systemic inflammation [14]. Moreover, cellular senescence is characterized by the activation of cell-cycle regulators, including either or both p53/p21^WAF1/CIP1^ and p16^INK4A^ [15]. Thus, targeting senescent cells is a promising strategy to alleviate or prevent aging and age-related diseases.

As part of this approach, researchers have been describing chemicals called ‘senolytics’ that cause the selective clearance of senescent cells by inducing apoptosis [16]. Recent studies suggest that senolytics can be expected to improve physical functions and extend health span as well as attenuate various age-related chronic disorders in aged animal models [17,18,19,20,21]. In some cases, these agents are derived from natural food resources, and consequently, have low toxicities [22].

Based on this conceptualization, we hypothesized that BG may provide a senolytic mimicking effect by regulating aging-related pathways due to its pleiotropic traits. Therefore, we examined the effects of BG on cellular senescence and regulation of age-related markers in vitro and in vivo, respectively.

## 2. Materials and Methods

### 2.1. Sample Preparations

Black ginseng (BG) extract was obtained from CJ CheilJedang Corporation (Suwon-si, Gyeonggi-do, Korea).

### 2.2. Animals and Experimental Design

All animals were housed at the National Institute on Aging (NIA) which is fully accredited by the American Association for Accreditation of Laboratory Animal Care. All animal procedures were approved by The Animal Care and Use Committee of the NIA Intramural Program. Young male C57BL/6J mice (9-week-old) were purchased from the Jackson Laboratory (Bar Harbor, ME, USA). Aged male C57BL/6 mice (18-month-old) were obtained from the National Institute on Aging (NIA) Aged Rodent Colony housed at Charles River Laboratories (Frederick, MD, USA). Animals were acclimated to the facility for 1 week with standard NIH chow (Teklad Global Rodent Diet, Envigo, Indianapolis, IN, USA) and water *ad libitum* after being transferred into the NIA intramural housing facility (Baltimore, MD, USA). They were next randomized into three groups: distilled water orally administered to 9-week-old mice (Young; n = 6), distilled water orally administered to 18-month-old mice (Old; n = 6), and 300 mg/kg BG administered to 18-month-old mice (Old + BG; n = 7). The dosage of BG was determined based on the previous study [23,24] After 4 weeks of oral gavage injection, mice were sacrificed to harvest liver, skeletal muscle (SKM), and white adipose tissue (WAT) for further analyses.

### 2.3. Primary Mouse Embryonic Fibroblasts (MEFs), Primary Hepatocytes Isolation and Human Embryonic Kidney (HEK) 293 Cell Cultures

Primary mouse embryonic fibroblasts (MEFs) were isolated from E13.5 embryos of 9-week-old C57BL/6J female mice (n = 3) or 8-week-old 129S1 female mice (n = 2). After the isolation, MEFs were maintained in high-glucose Dulbecco’s Modified Eagle Medium (DMEM) with sodium pyruvate containing 20% fetal bovine serum (FBS), 1% penicillin/streptomycin (P/S), 2 mM L-glutamine, non-essential amino acids (1×), and 20 mM HEPES. We isolated primary hepatocytes as previously described [25]. In brief, primary hepatocytes were isolated from C57BL/6J male mice (n = 3) and cultured in Waymouth’s media with 10% FBS, 1% P/S, 1 nM of insulin, and 1 uM of dexamethasone. HEK293 cells were maintained in high-glucose DMEM containing 10% FBS and 1% P/S. Cells were incubated for 3 days at 37 °C in 5% CO_2_.

### 2.4. Cellular Senescence Induction by Ionizing Radiation (IR) and Replicative Senescence (RS)

MEFs were seeded onto 60-mm culture plates and incubated to reach 70–80% confluency. Then, 5 μg/mL of aqueous BG extract was added to the cells for 7 days with media changes every 2 days. To determine the concentration of BG treatment, we treated it with BG in a dose-dependent manner as previously reported [26,27]. For acute irradiation, we decided on the ionizing radiation dose based on the previous study [26]. MEF cells with or without BG treatment were irradiated with a 20 Gy of γ-ray (^137^Cs source; Gammacell^®^ 40 Exactor, Best Theratronics Ltd., Kanata, ON, Canada). For RS study in a murine cell line, MEFs underwent cell division every 24–36 h for a few days until they reach senescence (i.e., passage 5). For RS study in a human cell line, we used HEK293 cells with over 30 passages. HEK293 cells were divided into two groups: vehicle (water) and BG treatment (5 ug/mL). Both groups underwent RS up to additional 10 passages from the initial.

### 2.5. Senescence-Associated β-Galactosidase (SA-β-gal) Staining

The irradiated cells were subjected to senescence-associated β-galactosidase (SA-β-gal) staining. SA-β-gal staining was conducted on MEFs using a Senescence β-Galactosidase Staining Kit (Cat No. #9860; Cell Signaling Technology, Inc., Danvers, MA, USA) according to the manufacturer’s instructions. The density of SA-β-gal-positive cells for each group was quantified using the ImageJ software provided by the National Institutes of Health (NIH). To quantify SA-β-gal-positive cells, the number of SA-β-gal-positive cells and the number of total cells were counted, and calculated the ratio of SA-β-gal-positive and total cells.

### 2.6. Real-Time Reverse-Transcription Polymerase Chain Reaction (RT-PCR)

Total RNA was extracted from individual frozen tissue samples using TRIzol Reagent (Invitrogen; Thermo Fisher Scientific, Waltham, MA, USA) and quantified using a NanoDrop OneC Microvolume UV-Vis Spectrophotometer (Thermo Fisher Scientific). The normalized RNA was reverse transcribed with an iScript™ Reverse Transcription Supermix for RT-qPCR (Bio-Rad Laboratories, Inc., Hercules, CA, USA). Gene expression was assessed using Power SYBR^®^ Green Master Mix (Applied Biosystems; Thermo Fisher Scientific) on a CFX Opus 384 Real-Time PCR System (Bio-Rad Laboratories, Inc.) with the following thermal cycling conditions: 95 °C for 10 min, followed by 39 cycles of 95 °C for 15 s and 60 °C for 1 min. The fluorescence cycle threshold value (Ct) data were normalized to 18S. The primer sequences are displayed in Appendix A.

For the assessment of the aging pathway, RT-PCR was conducted with RT^2^ Profiler™ PCR Array Mouse Aging (Cat No. PAMM-178Z; Qiagen, Hilden, Germany). In brief, total RNA from frozen liver tissue samples of Old and Old + BG groups was isolated by the Aurum™ Total RNA Mini Kit (Bio-Rad Laboratories, Inc.). The normalized cDNA, synthesized by iScript™ Reverse Transcription Supermix for RT-qPCR (Bio-Rad Laboratories, Inc.), was mixed with Power SYBR^®^ Green Master Mix (Thermo Fisher Scientific) and pipetted into the 384-well PCR array plates. The array was performed using a CFX Opus 384 Real-Time PCR System (Bio-Rad Laboratories, Inc.) with modified conditions according to the manufacturer’s instructions. The expression data were normalized to housekeeping genes on each RT^2^ Profiler PCR array plate (Actb, B2m, Gapdh, Gusb, and Hsp90ab1) and expressed for the control group. The fold changes of tested genes were determined by the 2^−ΔΔCT^ method with the online GeneGlobe Data Analysis Center provided by Qiagen (Hilden, Germany). The differences in mRNA expression between the Old and Old + BG groups were presented as a fold upregulation (+) or downregulation (−).

### 2.7. Western Blot Analysis

Cells were lysed by radioimmunoprecipitation assay (RIPA) buffer (Thermo Fischer Scientific, Waltham, MA, USA) containing phosphatase inhibitor (Phostop, Roche, Basel, Switzerland) and protease inhibitor cocktails (cOmplete, Roche, Basel, Switzerland). Tissue samples were homogenized with T-PER™ Tissue Protein Extraction Reagent (Thermo Fisher Scientific) containing phosphatase inhibitor and protease inhibitor cocktails via the OMNI BeadRuptor 12 (Omni-Inc, Kennesaw, GA, USA). Proteins were quantified using BCA Assay (Thermo Fisher Scientific) and then protein loading samples were resolved in SDS-PAGE under reducing conditions and transferred to polyvinylidene fluoride (PVDF) membrane. Membranes were blocked in blocking reagent (LI-COR, Lincoln, NE, USA) for 1 h at room temperature and incubated with primary antibodies overnight at 4 °C as follows: β-actin, β-catenin, cyclophilin B, GAPDH, p21, and p53 from Cell Signaling Technology, Inc. (Danvers, MA, USA), and p16 from Santa Cruz Biotechnology, Inc. (Dallas, TX, USA). After washing with Tris-buffered saline and Tween 20 (TBS-T) buffer 3 times for 12 min, membranes were incubated for 1 h in the appropriate secondary antibody added in antibody diluent at room temperature. Membranes were washed again 3 times with TBS-T and developed using a chemiluminescence assay system. Images on x-ray film were scanned, saved as TIFF files, and inverted, and integrated density was analyzed using ImageJ software.

### 2.8. Statistical Analysis

All experimental data are expressed as the mean ± standard error of the mean (SEM). Quantification analyses for western blot band density among three groups were conducted using one-way analysis of variance (ANOVA) followed by Tukey’s multiple comparisons. Student *t*-test was used for real-time PCR Ct values and immunoblotting between two groups. Statistical analyses were performed by Prism 9 software (GraphPad Software, San Diego, CA, USA).

## 3. Results

### 3.1. Black Ginseng Delays Cellular Senescence on MEFs

To examine whether BG could alleviate cellular senescence in vitro levels, we induced cellular senescence by exposing primary MEFs to acute IR. Senescence was detected using a senescence-associated beta-galactosidase (SA-β-gal) staining assay. We found that senescence was delayed in cells grown in media supplemented with 5 μg/mL of BG, compared to the non-supplemented control cells (Figure 1A). The distributions of SA-β-gal-positive cells for each group were digitized and quantified by ImageJ software (Figure 1B). These results imply that BG ameliorates cellular senescence by reducing the amount of IR-induced senescent cells. We observed a significant decrease in the percentage (%) of SA-β-Gal positive cells with 20 Gy + BG (*p* < 0.0001) (Figure 1C). To confirm BG’s effects on senescence, we conducted replicative senescence studies using two different cell lines, primary MEFs and HEK 293 cells. We kept them in culture until cells reached RS with or without BG treatment, then performed immunoblotting to evaluate a representative senescence marker, p53, between control and BG treated cells. We found that BG treatment during inducing RS reduced p53 expression levels in both cell lines (Figure 1D,E and Appendix A).

### 3.2. Black Ginseng Alters Hepatic Gene Expression Profiles Related to Aging-Associated Pathways in Aged Mice

To investigate how BG alleviates induction of senescence, the expression of aging-associated genes (a total of 84 genes) was analyzed using an RT^2^ Profiler PCR array. We observed that BG supplementation altered expression levels of hepatic genes in aged mice compared to control mice (Figure 2A). Among those 84 genes, 7 genes were differentially expressed in the BG-supplemented mice (*p* < 0.05). These genes are mostly anti-inflammatory genes as shown in the heatmap (Figure 2B). In support of this, BG supplementation resulted in decreased hepatic mRNA expression levels of senescence-associated secretory phenotypes (SASPs). Among SASPs, BG-supplemented mice had significantly downregulated gene expression levels of matrix metallopeptidase 12 (*Mmp12*) (Figure 2C). Additionally, mRNA expression levels of chemokine ligand 2 (*Cxcl2*), and plasminogen activator inhibitor-1 (*Pai-1*) were suppressed in BG-supplemented mice (Old+BG) compared to the control (Old) mice (Figure 2C).

### 3.3. Black Ginseng Regulates the Complement System and a Component of Wnt Signaling in Aged Mouse Livers

Our next focus was on which pathways led to changes in hepatic gene profiling. Thus, we normalized each gene on the PCR array between Old and Old+BG groups using a scatter plot and identified upregulated and downregulated genes (Figure 3A). BG supplementation upregulated genes associated with the inflammatory response, complement component 4A (*C4a*, *p* < 0.01), glial fibrillary acidic protein (*Gfap*), neurodegeneration and synaptic transmission associated gene, sodium channel subunit beta-2 (*Scn2b*), proteostasis related gene, and Janus kinase and microtubule interacting protein 3 (*Jakmip3*). Furthermore, among nine aging-related genes downregulated in the BG-supplemented aged mice, significant *p*-values were attained from four genes: *C1qb*, *C1qc*, *Clu*, and *Lsm5* (*p* < 0.05). Meanwhile, the expression levels of *Cd14*, *Ltf*, and *Tlr2* were downregulated (≥2.1-fold) in BG-supplemented aged mice, but the *p*-values were not significant (Figure 3B). We then performed the Clustergram for non-supervised hierarchical clustering of the entire dataset. This cluster analysis revealed that BG supplementation regulated the complement system (Figure 3A,B). *C1qb* and *C1qc* genes are components of the complement system, and *S100a8* and *S100a9* genes are known to regulate the complement system [28,29], indicating that BG supplementation mediates the complement system (Figure 3B). To further identify biological pathways regulated by this system, we next examined the canonical Wnt signaling pathway because of its involvement in aging [30]. The complement C1q family activates Wnt signaling by binding to Frizzled receptors [31]. Among components of Wnt signaling, we primarily analyzed the β-catenin protein expression level, which is a pivotal component of Wnt signaling [32]. Of importance, BG supplementation significantly decreased hepatic β-catenin expression levels compared to the aged control mice (*p* < 0.05, Figure 3C and Appendix A).

### 3.4. Black Ginseng Supplementation Ameliorates Canonical Cellular Senescence Pathways in Metabolically Active Organs in Aged Mice

Previous studies have revealed that Wnt signaling is involved in aging mammalian phenotypes because it regulates cell fate and proliferation choices [32,33,34]. Deregulated β-catenin induces accumulation and activation of p53 tumor suppressor protein [35,36]. Therefore, we evaluated the expression of p53 and p53-dependent p21, both of which are involved in the canonical cellular senescence pathway and were decreased in primary hepatocytes from Old+BG mice compared to Old mice (Figure 4A). Moreover, another conventional p16 signaling pathway involved in mediating the senescence process [37] was downregulated in BG-supplemented aged mice (Figure 4A). These biomarkers, p53, p21, and p16, contribute to metabolic disease and determine the influence of senescent cells on metabolic activity [38,39,40]. We, therefore, investigated the effects of BG on the senescence process in other highly metabolically active organs, namely the liver, skeletal muscles (SKM), and white adipose tissues (WAT), and, consistent with results in the primary hepatocytes, the expression levels of p53, p21 and p16 showed similar trends (Figure 4B–D and Appendix A).

## 4. Discussion

Cellular senescence, irreversible cell-cycle arrest coupled with SASP, is a characteristic of aging. Therefore, a promising approach for delaying or preventing age- and cellular senescence-associated pathologies is to eliminate senescent cells. BG is renowned for its beneficial effects on factors that contribute to age-related declines in function because it has anti-inflammatory, anti-viral, anti-carcinogenic, and antioxidant effects when it is consumed [3,4,5,7,8,23]; however, it is still unknown how BG can alleviate cellular senescence at the molecular level.

This is the first study to demonstrate that BG reduces cellular senescence in MEF and influences canonical cellular senescence pathways in aged mice. Our results with IR-exposed MEF indicated that BG can protect IR-induced cellular senescence. The canonical senescence pathway, the p53-p21 signaling pathway, is the most studied mechanism to describe the underlying mechanisms of IR-induced cellular senescence [41]. One of the conserved pathways of cellular growth arrest is governed by p53 and p21, which play key roles in the initiation of senescence [42,43]. The other noted pathway is p16/pRB which has a central role in the maintenance of senescence [42,44]. Activated p53 promotes senescence by inducing p21, leading to G1 arrest of the cell cycle [12,15,16,45]. Regulated independently by p53, p16-mediated senescence controls G1 to S transition by preventing Rb phosphorylation [44,45,46]. Accumulation of these biomarkers, including the inflammatory cascade in metabolic organs—liver, SKM, and WAT—has a causal relationship with hepatic steatosis, cirrhosis, and sarcopenia [47,48,49]. In this study, we elucidated that BG supplementation downregulated p53 protein levels in major metabolic organs: liver, SKM, and WAT. Additionally, protein levels of the p53-dependent downstream regulator, p21, were decreased in BG-supplemented aged mice. These findings indicate that BG supplementation has the potential to be a senolytic agent that would ameliorate metabolic dysfunction since cellular senescence occurrence in highly metabolic organs is a feature of age-related metabolic diseases [39,40]. Furthermore, BG supplementation could alleviate metabolic dysfunction and lessen the chance of hepatic steatosis, cirrhosis, and sarcopenia by maintaining tissue homeostasis. There is support for this possibility because BG supplementation mediates the p16 pathway. However, p21 prevents the inactivation of pRB, a marker of cellular and replicative senescence, which is also controlled by p16 in senescence pathways [46,50]. In this study, our results in WAT implicated that the level of p21 and p16 are variable within the groups. Adipose tissue consists of different types of cells such as proliferative and postmitotic cells, and their roles in senescence are distinguished. Unlike p21, p16 is activated in most cells in WAT during cellular senescence [51,52,53]. For this reason, we need to further study whether BG supplementation regulates the p16 pathway in a p53-p21-dependent or independent manner in WAT.

Previous studies have reported that BG reduces ROS production, resulting in less secretion of inflammatory cytokines and downregulation of the inflammatory cascade [7,23,24]. These studies focused on BG’s effects on the intrinsic markers of oxidative stress and inflammation, whereas we studied its effects on the age-associated inflammatory response-associated hepatic transcriptome profile. Moreover, we conducted a mechanistic study to uncover if there is a link between altered inflammatory response and prevention of cellular senescence. We observed that BG supplementation primarily regulates the complement system. The complement system plays a critical role in innate immunosurveillance by defending against common pathogens. The complement system consists of more than 30 plasma proteins produced mainly by the liver. These proteins construct a cascade for opsonization and induction of inflammatory response [54,55]. Complement C1q, a pattern recognition molecule, is the first recognition molecule of the complement classical pathway, which is intrinsic to the innate immune system [56,57]. Recently, studies have reported C1q as a multi-functional molecule in homeostasis and development that is irrelevant to the classical pathway activation [58]. Of importance, this molecule plays a critical role in aging by impairing the regenerative capacity in various tissues by activating the canonical Wnt signaling pathway [31]. In the same context, we found that BG supplementation suppressed C1q expression and inhibited Wnt signaling in this study.

BG is a heterogeneous mixture of pharmacological saponin glycosides and ginsenosides and has abundant amounts of protopanaxadiol-type saponins (e.g., Rg3, Rk1, and Rg5) and protopanaxtriol (Rh1) [5]. Previously, it was shown that Rg5 prevents hepatic oxidative stress and heat-induced inflammation [24]. Additionally, ginsenosides Rg3 and Rh1 suppress inducible nitric-oxide synthase and the secretion of inflammatory cytokines [59,60]. However, we cannot rule out the anti-inflammatory functions of the rare ginsenosides (Rg2, Rg4, Rg6, and Rh4) in the whole BG extract [61]. In the present study, we used the whole extract of BG for in vitro and in vivo studies and demonstrated that BG supplementation results in diminished cellular senescence. Considering the use of BG as a dietary supplement, evaluation of the whole extract’s function is a more practical approach than the use of a specific ginsenoside. However, we still need to determine which type(s) of ginsenoside is the most powerful bioactive senolytics in BG.

There are a few limitations to this study. Although BG showed decreased percentages (%) of SA-β-Gal positive cells, an in-depth mechanistic explanation of how BG regulates canonical senescence pathways in IR-induced MEFs and multiple cell lines is needed. In the current study, we demonstrated the preventive effects of BG on canonical senescent pathways, p53-p21/p16. However, further study of these observations is needed in the metabolic organs for age-associated diseases such as nonalcoholic fatty liver disease (NAFLD), sarcopenia, and geriatric diabetes.

## 5. Conclusions

Our research demonstrates that BG supplementation of in vitro and aged mice attenuate cellular senescence by downregulating complement C1q and β-catenin signaling and their downstream activators in senescent pathways: p53-p21/p16. Taken together, our results suggest that BG could be a novel senolytic by postponing cellular senescence.

## Figures and Tables

**Figure 1 biology-11-01108-f001:**
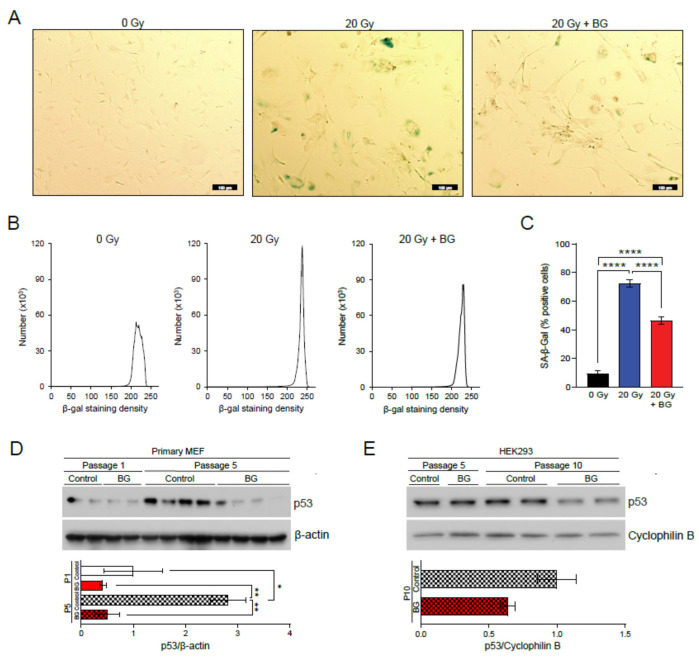
Representative morphologies of senescence-associated β-galactosidase (β-gal) stained (bluish-green color) primary mouse embryonic fibroblasts (MEFs) with 0 Gy, 20 Gy, 20 Gy+BG (5 μg/mL) (scale bar = 100 μm) (**A**), SA-β-gal quantification plots based on staining density (**B**), and SA-β-gal quantification plots based on percentage of positive cells (**C**), p53 protein expression levels in primary MEF at passage 1 (n = 2 per group) and passage 5 (n = 4 per group) with control and BG treatment (5 ug/mL) (**D**), p53 protein expression levels in human embryonic kidney (HEK) 293 cells at passage 5 and 10 (n = 1–2 per group) with control and BG treatment (**E**) * *p* ≤ 0.05, ** *p* ≤ 0.01 and **** *p* ≤ 0.0001.

**Figure 2 biology-11-01108-f002:**
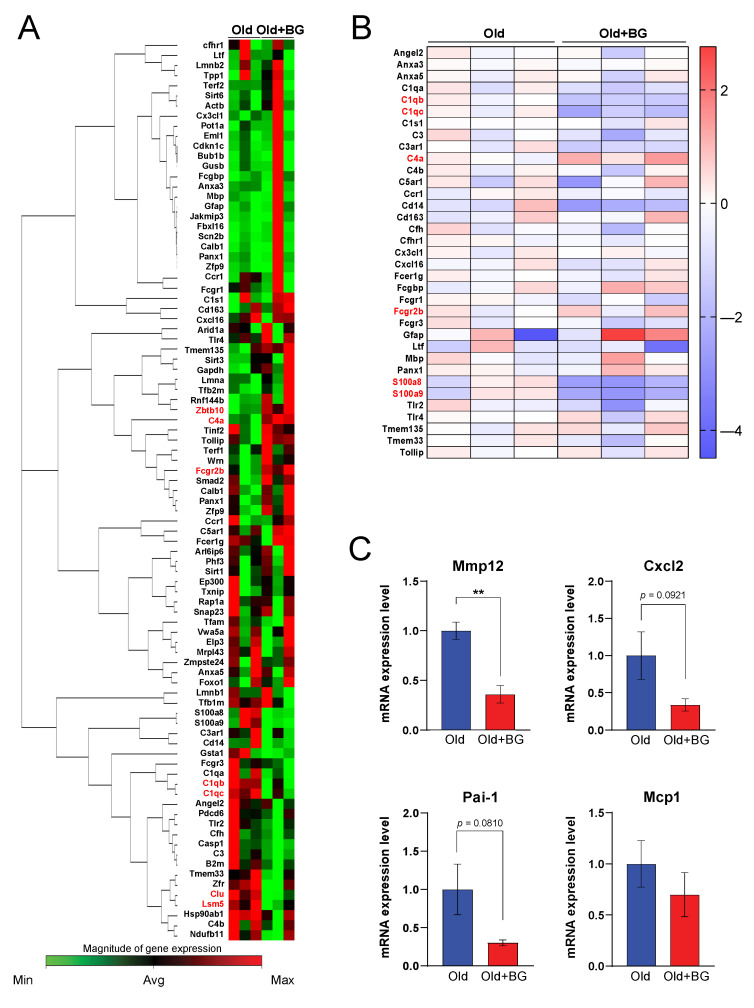
Clustergram (**A**) of aging-associated genes, heat map (**B**) of inflammation-related genes, and mRNA expression level (**C**) of SASP factor genes (*Mmp12, Cxcl2, Pai-1, Mcp1*) in liver isolated from Old or Old+BG mice (n = 3). ** *p* ≤ 0.01.

**Figure 3 biology-11-01108-f003:**
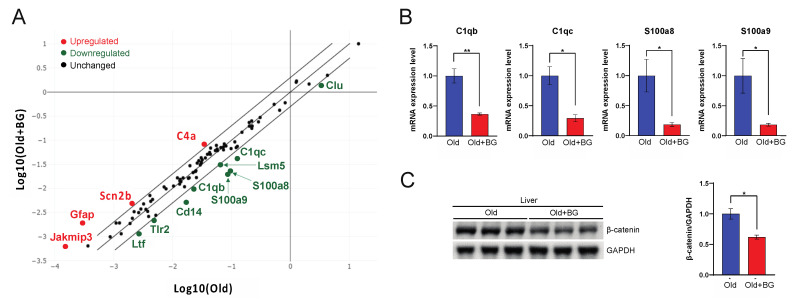
Scatter plot (**A**) of aging-associated genes between Old (*x*-axis) and Old+BG (*y*-axis) group, an mRNA expression level (**B**) of *C1qb*, *C1qc*, *S100a8*, *S100a9,* and western blot (**C**) of β-catenin and its quantification. In Figure 3A, the red, green, and black dots indicate upregulated, downregulated, and unchanged genes, respectively. * *p* ≤ 0.05 and ** *p* ≤ 0.01.

**Figure 4 biology-11-01108-f004:**
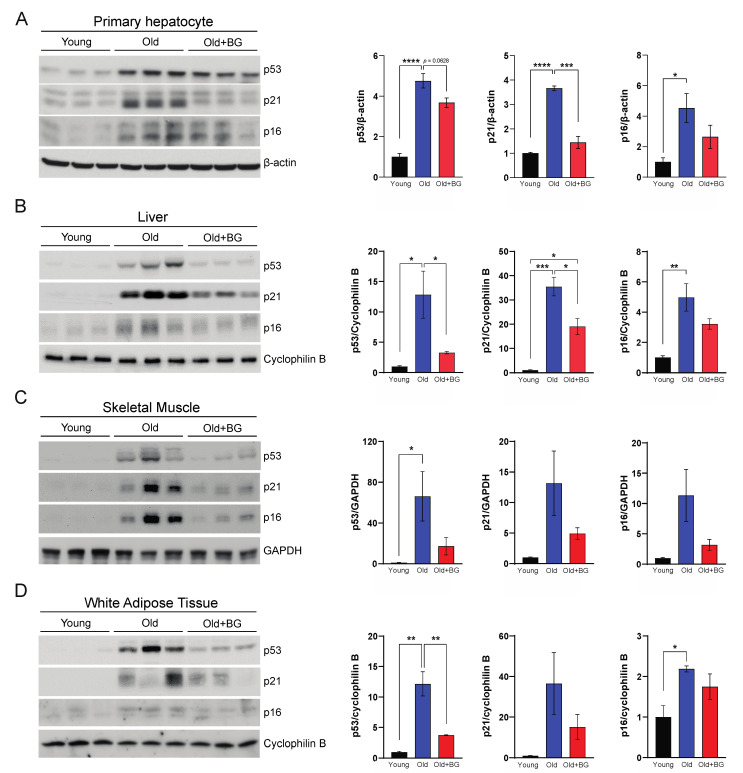
p53, p21, and p16 expression level in primary hepatocytes (**A**), liver (**B**), skeletal muscle (SKM) (**C**), and white adipose tissue (WAT) (**D**) (n = 3 mice per group). * *p* ≤ 0.05, ** *p* ≤ 0.01, *** *p* ≤ 0.001 and **** *p* ≤ 0.0001.

## Data Availability

Not applicable.

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
