# Peer review of "Black Ginseng Ameliorates Cellular Senescence via p53-p21/p16 Pathway in Aged Mice"

_biology, 2022, doi:10.3390/biology11081108_

Round 1

Reviewer 1 Report

Lee et al. study the role of Black ginseng in alleviating deleterious effects of cellular senescence. Overall the concept is exciting, and if it works very efficiently, it will be beneficial for aging research. Overall the manuscript needs a lot of work to strengthen the authors' claims. I have substantial inhibitions for this manuscript to publish with this data. My comments pointwise below:

  • The manuscript needs better resolution figures, especially figure-2, which is illegible.
  • What is the final radiation dose used for irradiation experiments?
  • Characterization of senescent MEF has not been done. SAbeta gal alone is not a good measure of senescence.
  • Figure 1 is unclear, SAbeta gal is not obvious, and the quantitation is not ideal. Please check the other papers that use SAbeta gal as a marker for senescence. They generally use percentage SAbeta gal positive cells in the total cells.
  • In figure 4, why are few graphs normalized to GAPDH and few to cyclophilin B?
  • In figure 4, why do the mice have variable levels of p53?
  • The authors mention that BG acts as a senolytics. Which data shows that it acts as senolytics or the death of senescent cells?
  • The authors used only one cell line to show the senolytic effects of BG. Do they see the same effect in any human or other mouse cells?
  • Generally, senolytics positively affect aging illnesses in old mice, such as decreased sarcopenia, kyphosis, etc.   What is the phenotypic effect of BG on old mice?

Reviewer 2 Report

In this study, Su-Jeong Lee et al demonstrated that black ginseng BG positively impacts the age-related phenotypes by affecting expression of cellular senescence p53-p21/p16 pathway in the liver and other metabolic organs of aged mice. In general, this topic is interesting for a broad readership, dealing with the molecular mechanisms of black ginseng affecting cellular senescence. However, there are still many limitations affecting conclusion and significance.

In details:

  1. The anti-aging effects of ginseng have been extensively reported in the literature. As one of the processed ginseng products, several studies have revealed that BG has ameliorative effects on age-related functional decline. So please point out the novelty of your present study.
  2. In vitro experiment, 5μg/mL BG delayed cellular senescence on MEFs. How to determine the concentration of BG? No relative experiment or the literature cited. Whether 5μg/mL BG has effects on the growth of MEFs? In addition, for ionizing radiation, cells were irradiated with a 20 Gy/min of γ-ray. The radiation time also needs to be explained.
  3. The authors only detected SA-β-gal staining to determine the cellular senescence in vitro experiment, it is completely adequate. P53-p21/p16 pathway should be investigated to verify the effects of BG supplementation on MEFs.
  4. BG supplementation in aged mice significantly inhibited the activation of the classic pathway of aging, p53-p21/p16. However, in-depth mechanistic exploration is necessary.
  5. Besides protein expression levels of p53, p21 and p16 for cellular senescence, it is recommended to add assays for mRNA expression.
  6. There are several major discrepancies on results described in the text vs. those shown in Figures. For example, text in Results stated that “the expression levels of Cd14, Ltf, S100a8, S100a9, and Tlr2 196 were downregulated (≥ 2.1-fold) in BG supplemented aged mice, but the p-values were not significant”. However, p valves of S100a8 and S100a9 in Figure 3B were both less than 0.05. The authors should clarify the discrepancies and check on the text statements against figures. Moreover, why C1qb, C1qc, S100a8 and S100a9 among nine aging-related genes were selected for demonstration in Figure 3B?
  7. In Figure 1, SA-β-gal positive rate should be analyzed, and whether there are differences among the three groups.
  8. In your supplementary material, Figure 3D should be changed to Figure 3C.
  9. Although English language is readable, it still requires wide fine tuning. 

Reviewer 3 Report

The manuscript by Lee et al explores the therapeutic role of black ginseng (BG) in alleviation of senescence phenotypes. The authors compared gene expression profile for 84 genes, associated with aging, among old and old +BG treated mice and found that BG treatment promoted youthful expression levels of some genes relative to the control. Using a combination of in vitro and in vivo analysis, the authors found that BG supplementation suppressed C1q expression, which is an inducer of pro-aging Wnt signaling. In support of this, the Wnt signaling and other biomarkers of senescence were shown to be downregulated in BG treated mice relative to the control. The study for most part is well carried out and the the manuscript is well written. There is however, some disconnect in how some of the data is described and discussed. The manuscript will benefit from addressing the following points prior to publication.

Major points-

  1. The authors should test the efficacy of BG on more than one senescence inducing agents.
  2. Line 170: There is a disconnect in first showing senescence in MEFs (which have not been aged), and then immediately following up on the involvement of aging-associated genes in BG-dependent alleviation of senescence in tissue-specific manner. The authors should also measure senescence in young, old, and old + BG treated mice to establish that the changes in gene expression and metabolic activities being observed in these groups correlate with their senescence levels. This will also help to establish if the percent rescue of senescence by BG in vitro is similar to the percent rescue in vivo.
  3. The authors should compare the expression patterns of the 84 genes between young, old, and old + BG treated. This will be more informative in concluding if BG treatment indeed helps in reversing aging-associated gene expression patterns and by how much.
  4. Some background should be provided on the complement system.
  5. The blot in 3C should also include samples from young mice.
  6. The immunoblot data from white adipose tissue has more variability than other tissues. Between old and old + BG, p21 blot has all possible scenarios- no change, increase, and decrease. And p16 blot appears more or less consistent across all groups. This variability in white adipose tissue data should be acknowledged in the text.
  7. Since the authors have looked at metabolically active tissues, it will be worthwhile to discuss the role BG treatment might have on the metabolic state of these tissues, if improving metabolic flexibility might also be a part of the mechanism by which BG exerts its anti-aging benefits.

Minor points:

  1. Figure legends should be thoroughly described. For example, in 1A, what does 20 reflect? In 3A, what do the axes represent in the scatter plot? Why are the values in the negative quadrant? How was the cutoff defined for the upregulated and downregulated genes? The legend should describe what each of the colors and arrows mean.
  2. Highlight the 7 differentially expressed genes in 2A and 2B.
  3. No rationale has been provided on how and why the genes shown in 2B were selected to build another heatmap.
  4. Error bars are missing for some graphs in figure 4.

Reviewer 4 Report

The study was aimed at evaluating the effect of Black Ginseng in aging processes. Using both in vitro and in vivo approaches the authors demonstrated the capability of this compound to downregulate complement C1q and β-catenin signaling as well as their downstream activators p53 and p21/p16. The results support the idea of BG as novel senolytic agent. The manuscript is well written, subject matter original and of major general interest. Procedures are clear and easily replicable. The results are logically presented and stated statistical significance of findings. The discussion is focused on the data and conclusions are linked to goals. However, study limitations should be added.

Figures contribute substantially to content. References correlated well with the text

Author Response

The study was aimed at evaluating the effect of Black Ginseng in aging processes. Using both in vitro and in vivo approaches the authors demonstrated the capability of this compound to downregulate complement C1q and β-catenin signaling as well as their downstream activators p53 and p21/p16. The results support the idea of BG as novel senolytic agent. The manuscript is well written, subject matter original and of major general interest. Procedures are clear and easily replicable. The results are logically presented and stated statistical significance of findings. The discussion is focused on the data and conclusions are linked to goals. However, study limitations should be added.

Figures contribute substantially to content. References correlated well with the text.

We added sentences to provide study limitations in Discussion (line 301-307)

  • Lines (301-307): There are a few limitations in this study. Although BG showed decreased percentages (%) of SA-β-Gal positive cells, an in-depth mechanistic explanation on how BG regulates canonical senescence pathways in IR-induced MEFs and multiple cell lines is needed. In the current study, we demonstrated the preventive effects of BG on canonical senescent pathways, p53-p21/p16. However, further study of these observations is needed in the metabolic organs for age-associated diseases such as nonalcoholic fatty liver disease (NAFLD), sarcopenia and geriatric diabetes.

Round 2

Reviewer 1 Report

Thanks for the revised version of the manuscript, "Black ginseng ameliorates cellular senescence via p53-p21/p16 2 pathway in aged mice".

I am very happy to see that the authors have added a new and high-resolution figure 2, and it is much clear. Even though the figure is better, the main takeaway from the figure is still missing. What about the other markers associated with senescence, such as IL-6, IL-1B, and other markers such as Lamin B1, which negatively correlate with senescence?

The SAbeta gal labeling is still very clear. Other markers such as p16 or p21 can be used to do antibody labeling to show senescence.

The data is not supported by the in-vivo amelioration of the senescence and aging phenotypes with Black ginseng.

Authors need a lot of work before it can be published. In my opinion, it still needs a lot of data to support the claims authors are trying to make. The meaningful data is still not present in the manuscript that is needed for publication. 

Author Response

Thanks for the revised version of the manuscript, "Black ginseng ameliorates cellular senescence via p53-p21/p16 2 pathway in aged mice".

I am very happy to see that the authors have added a new and high-resolution figure 2, and it is much clear. Even though the figure is better, the main takeaway from the figure is still missing. What about the other markers associated with senescence, such as IL-6, IL-1B, and other markers such as Lamin B1, which negatively correlate with senescence?

We measured IL-6 and IL-1b mRNA expression levels in the liver. For IL-1b, the levels are Young vs Old vs Old+BG (1±0.228 vs 2.11±0.673 vs 0.87±0.279, mean ± SEM, n=4/group). Although the data show a negative correlation with senescence, the Old group’s SEM is quite big. Thus, it did not reach significance. To avoid misunderstanding for readers, we did not report the data in the current manuscript. Also, we performed SYBR green qPCR for several inflammatory senescence-related genes such as IL-6 and IFN-γ, but they were detected over 36 cycles with high variations. Even a few liver samples from the Young group were not detected within 40 cycles. We plan to examine more senescence markers in order to show BG’s negative correlation with senescence in the follow-up study.

The SAbeta gal labeling is still very clear. Other markers such as p16 or p21 can be used to do antibody labeling to show senescence.

We appreciate your suggestion about using a method with antibody labeling of other markers such as p16 or p21. However, we cannot currently access the same irradiator for the primary MEF study. Also, primary MEF studies take more than 2 months because we isolate primary MEFs from C57Bl/6 female mice who have been timed mated to obtain embryonic day (E13.5). We are sorry that we have been unable to do this experiment at this point. Instead, we added sentences to provide a more clear rationale as to why we used primary MEFs in the Results (lines 160-161) and our future studies in the Discussion (lines 301-303)

  • Lines (160-161): To test our hypothesis that BG may alleviate cellular senescence, we induced cellular senescence by exposing MEFs to acute ionizing radiation (IR).

  • Lines (301-303): There are a few limitations in this study. Although BG showed decreased percentages (%) of SA-β-Gal positive cells, an in-depth mechanistic explanation on how BG regulates canonical senescence pathways in IR-induced MEFs and multiple cell lines is needed

The data is not supported by the in-vivo amelioration of the senescence and aging phenotypes with Black ginseng.

We are planning to conduct a follow-up study to apply our findings for prevention of aging phenotypes and age-associated diseases such as nonalcoholic fatty liver disease (NAFLD), sarcopenia and geriatric diabetes. We added these sentences in the Discussion (lines 303-307).

  • Lines (303-307): In the current study, we demonstrated the preventive effects of BG on canonical senescent pathways, p53-p21/p16. However, further study of these observations is needed in metabolic organs for age-associated diseases such as nonalcoholic fatty liver disease (NAFLD), sarcopenia and geriatric diabetes.

Authors need a lot of work before it can be published. In my opinion, it still needs a lot of data to support the claims authors are trying to make. The meaningful data is still not present in the manuscript that is needed for publication. 

Reviewer 2 Report

The authors have addressed some of my concerns. However, there are still many limitations affecting conclusion and significance.

  1. In vitro and in vivo experiments in the present manuscript can’t be combined well to illustrate the study. For example, in vitro experiment, MEF senescence induced by ionizing radiation were used to explore the effects of BG, and there was no relevant mechanistic exploration. However, in vivo experiment, aged mice, metabolically active organs and primary hepatocytes were investigated.
  2. Although the authors added the mechanistic explanation in Discussion, there was no experimental data to support it. In-depth mechanistic exploration is necessary.
  3. Why C1qb, C1qc, S100a8 and S100a9 among nine aging-related genes were selected for demonstration in Figure 3B? The author did not clarify.

Author Response

The authors have addressed some of my concerns. However, there are still many limitations affecting conclusion and significance.

  1. In vitro and in vivo experiments in the present manuscript can’t be combined well to illustrate the study. For example, in vitro experiment, MEF senescence induced by ionizing radiation were used to explore the effects of BG, and there was no relevant mechanistic exploration. However, in vivo experiment, aged mice, metabolically active organs and primary hepatocytes were investigated.

In this study, we reported SA-β-gal staining data using senescence-induced primary MEFs for preliminary examination of whether BG treatment affects the senescence process. The central hypothesis of this study is ‘BG supplementation prevents or delays cellular senescence in aged mice’. Also, the rationale for using primary MEFs is that we used an aged C57Bl/6 mouse model in this study. To support the results from the same mouse strain, we isolated primary MEFs from the same mouse strain, C57Bl/6, and induced senescence instead of using other commercially available cell lines. In our study, we consistently used primary MEFs, primary hepatocytes, liver, skeletal muscle and white adipose tissue from the same mouse stain.

Also, we added sentences to provide more clear rationale why we used primary MEFs study in Result (lines 160-161).

  • Lines (160-161): To test our hypothesis that BG may alleviate cellular senescence, we induced cellular senescence by exposing MEFs to acute ionizing radiation (IR).

  1. Although the authors added the mechanistic explanation in Discussion, there was no experimental data to support it. In-depth mechanistic exploration is necessary.

We added sentences to provide study limitation in the Discussion (line 301-307).

  • Lines (301-307): There are a few limitations in this study. Although BG showed decreased percentages (%) of SA-β-Gal positive cells, an in-depth mechanistic explanation on how BG regulates canonical senescence pathways in IR-induced MEFs and multiple cell lines is needed. In the current study, we demonstrated the preventive effects of BG on canonical senescent pathways, p53-p21/p16. However, further study of these observations is needed in the metabolic organs for age-associated diseases such as nonalcoholic fatty liver disease (NAFLD), sarcopenia and geriatric diabetes.

  1. Why C1qb, C1qc, S100a8 and S100a9 among nine aging-related genes were selected for demonstration in Figure 3B? The author did not clarify.

We revised the manuscript in order to make this clear (line 207-209).

  • Lines (207-209): C1qb and C1qc genes are components of the complement system, and S100a8 and S100a9 genes are known to regulate the complement system [27,28], indicating that BG supplementation mediates the complement system (Figure 3B).

Reviewer 3 Report

The authors have adequately addressed the concerns I raised. If the journal word limit permits, the clarifications that the authors have provided for major points 1 and 2 can also be incorporated into the manuscript. This will strengthen their rationale and will add clarity for the reader.

Author Response

The authors have adequately addressed the concerns I raised. If the journal word limit permits, the clarifications that the authors have provided for major points 1 and 2 can also be incorporated into the manuscript. This will strengthen their rationale and will add clarity for the reader.

We added sentences of study limitation in the Discussion (line 301-307)

  • Lines (301-307): There are a few limitations in this study. Although BG showed decreased percentages (%) of SA-β-Gal positive cells, an in-depth mechanistic explanation on how BG regulates canonical senescence pathways in IR-induced MEFs and multiple cell lines is needed. In the current study, we demonstrated the preventive effects of BG on canonical senescent pathways, p53-p21/p16. However, further study of these observations is needed in the metabolic organs for age-associated diseases such as nonalcoholic fatty liver disease (NAFLD), sarcopenia and geriatric diabetes.

Round 3

Reviewer 2 Report

The authors addressed most of the comments and submitted an improved manuscript to the journal.

Author Response

We appreciate your time and effort to review our manuscript.